# School Health Services and Health Education Curricula in Greece: Scoping Review and Policy Plan

**DOI:** 10.3390/healthcare11121678

**Published:** 2023-06-07

**Authors:** Pelagia Soultatou, Stamatis Vardaros, Pantelis G. Bagos

**Affiliations:** 1Department of Public and Community Health, University of West Attica, 11521 Athens, Greece; 2Department of Political Science, University of Crete, 74100 Rethymno, Greece; st.vardaros@gmail.com; 3Department of Computer Science and Biomedical Informatics, University of Thessaly, 35100 Lamia, Greece

**Keywords:** public health, policy plan, school health, health education, youth

## Abstract

The new generation’s health and wellbeing is of paramount importance: it constitutes United Nations’ priority, complies with Children’s Rights and responds to the Sustainable Development Goals of the United Nations. In this perspective, school health and health education, as facets of the public health domain targeted at young people, deserve further attention after the unprecedented COVID-19 pandemic crisis in order to revise policies. The key objectives of this article are (a) to review the evidence generated over a span of two decades (2003–2023), identifying the main policy gaps by taking Greece as a case study, and (b) to provide a concrete and integrated policy plan. Following the qualitative research paradigm, a scoping review is used to identify policy gaps in school health services (SHS) and school health education curricula (SHEC). Data are extracted from four databases: Scopus, PubMed, Web of Science and Google Scholar, while the findings are categorized into the following themes following specific inclusion and exclusion criteria: school health services, school health education curricula, school nursing, all with reference to Greece. A corpus of 162 out 282 documents in English and Greek initially accumulated, is finally used. The 162 documents consisted of seven doctoral theses, four legislative texts, 27 conference proceedings, 117 publications in journals and seven syllabuses. Out of the 162 documents, only 17 correspond to the set of research questions. The findings suggest that school health services are not school-based but a function of the primary health care system, whereas health education retains a constantly changing position in school curricula, and several deficiencies in schoolteachers’ training, coordination and leadership impede the implementation. Regarding the second objective of this article, a set of policy measures is provided in terms of a problem-solving perspective, towards the reform and integration of school health with health education.

## 1. Introduction

The unprecedented crisis of the COVID-19 pandemic provoked a discussion in respect to public health policies fueled by the need to respond to the wide spectrum of emerging challenges aligned to the plethora of exacerbated vulnerabilities and inequalities evidenced through the humanitarian crisis, as a growing corpus of literature shows [1,2,3,4]. The imperative to invent a new vision for public health which addresses the challenges, takes seriously into account the lessons emerging from the humanitarian crisis and reorganizes public health upon the ideal of a common good is also echoed in the literature [5,6,7]. However, little attention has been paid to school health services (SHS) and school health education curricula (SHEC), as two inter-linked dimensions of public health aiming at youth.

The new generation’s health and wellbeing is of paramount importance, constitutes United Nations’ priority and complies with Children’s Rights [8]. Seventeen Sustainable Development Goals (SDG) have been adopted by governments at the UN General Assembly relating the SDG3 directly to health, prompting the states to ensure healthy lives and promote well-being for all at all ages. The SDG declaration emphasizes that to achieve the overall health goal, universal health coverage and access to quality health care will be encompassed in health policies. The UN Convention on the Rights of the Child entails that states need to ensure institutions, services and facilities responsible for the care and protection of children will conform with the standards established by competent authorities, in respect to health [9]. It is advocated to work towards “a revitalized global effort to fully protect, nurture, and support the health and development potential for every child everywhere, from before conception to adulthood” [10] (p. 1761).

Primary health care has the potential to fulfill the SDG in line with the Declaration of Astana, which defines three key areas of primary health care as: service provision, multi-sectoral actions and the empowerment of citizens. It also provides the framework to achieve the ideal of universal health coverage as it may prevent disease, promote health, reduce growth in costs and inequality, if the core primary health care principles (outlined as first contact, continuous, comprehensive and coordinated care) are translated into practice [11]. In terms of public health crises, it was found that countries that offer either universal medical care or universal health insurance systems ensured a better response to the disastrous effect of the pandemic on the most at-risk populations [12]. However, the provision of PCH and the achievement of the SDGs may be impeded by inadequate government spending on health, the shortage and maldistribution of the health workforce and an inadequate multi-sectoral health workforce [13,14].

School health services may be provided by health professionals and allied professionals either employed in local centers of the primary health care system or other public health units. SHS have the potential to address health inequalities for vulnerable children in deprived regions and lower socioeconomic strata and provide in situ health services and effective interventions fulfilling the goals of disease prevention and health promotion. However, as a recent large-scale study documented though a comparative study of 30 European countries, substantial disparities exist between countries as to the provision of services and most of the countries under consideration report a shortage in school health professionals, whilst training in school health is deficient [15].

School health education is aligned with SHS in that it constitutes the essentially pedagogical tenet of public health traced basically in school setting. SHE may adopt the values of neoliberal public health policies representing public health crises as unique and accidental, concurrently emphasizing individual responsibility, and adopting a victim-blaming perspective [16]. A new curriculum needs to be devised based on the ideals of democracy, social justice and solidarity and adopting a whole-school, bottom-up, pupil-centered and collaborative methodology. A paradigm shift from the individualistic and victim-blaming conception of health which is a product of the neoliberal ideas’ dominance [17] will take seriously into account the lessons derived from this unprecedented health crisis which exacerbated social inequalities.

The pandemic health crisis is arguably not isolated from the wider crises of the capitalist system since the root causes of the pandemic include such things as “capitalist agriculture, its destruction of natural habitat, and the industrial production of meat” [18] (p. 55). School health education is founded on the principles of equality and equity, and incorporating also an ecological dimension will serve this purpose. For instance, it is suggested that the health promotion agenda should be reframed, based on three aspects: (a) planetary issues, (b) governance and (c) civil society and social change [19]. Similarly, the “5 Ps model” of global health education, which refers to parity, people, planet, priorities and practices, where parity stands for health equity, may be taken as an example of how the new health promotion agendas, imbued with humanitarian values, may expand to planetary issues, viewing individual and community health as inseparable from climate change [20].

It is evident from the above that in the post-pandemic world the new vision of public health services with emphasis on youth should not be viewed in isolation from the persistent and overlapping capitalist crises and the dismantling effects of neoliberal policies against state capabilities in favor of the markets. Quite the contrary, we argue that health inequalities, especially the exacerbated vulnerabilities of young people, may be alleviated if a radical and substantive change is planned and implemented in the post-pandemic world. This change needs to take the social determinants discourse seriously but also to include several aspects, such as the economy, the environment, social protection labor markets, education and skills formation as a public good, and an international obligation is proposed.

## 2. The Context of This Study

School health and health education constitute distinct responsibilities between two public executive bodies, the Hellenic Ministry of Health, and the Ministry of Education. This creates space for dichotomy, disintegration and lack of coordination, which therefore may lead to an expansion of unmet health needs, an indicator that signals social inequalities in health and in which Greece lags behind significantly [21]. Second, Greece has suffered the effects of a prolonged and severe economic crisis for nearly a decade (2009–2018) which was then succeeded by the multitude of capitalist crises provoked by the pandemic of COVID-19, which also impact on the supply of the health system and the educational system.

First, school health and health education, as interlinked fields of theory and practice, constitute a shared institutional responsibility between, respectively, the Ministry of Health (MoH) and the Ministry of Education (MoE). The MoH is responsible for delivering health services and health education programs in the school setting, whereas the MoE is accountable for educational policies that will encompass elements of health education. To date, there is no legislation or memorandum of understanding between the two executive bodies towards a common framework of policy planning. Within this framework, the health policy, represented by school health services, and the educational policy, represented by school health education, are deficient in respect of: (a) a common understanding of what children’s and adolescents’ health entails conceptually and how health needs may be met methodologically and (b) a concrete partnership and coordination mechanism between MoH and MoE to translate theory into practice [22].

Second, school health services are offered by Local Primary Care Units (TOMY) for urban regions units according to the legislative framework (law 4486/2017) that sought to reform and reinforce Primary Health Care (PHC). However, in practical terms, this is rarely the case due to TOMYs’ limited capacity due to the prolonged period of austerity measures and neoliberal policies. Moreover, PHC services have been for long devalued as the health care system has been strongly hospital-oriented [10] and it has been characterized by lack of integration, fragmentation, shortcomings in efficiency and inequalities in access [23,24]. However, access to PHC through a multidisciplinary team was intended to ensure universal coverage [13]. The legal framework was enacted within months of the announcement. However, it encountered severe problems such as the significantly low availability of general practitioners and pediatricians, who resisted the idea of working exclusively for PHC [14], while it also met the undermining critique of organized economic interests and the neoliberal political party of New Democracy which was profoundly against the idea of expanding the social state. Under the circumstances, from 2017 to 2019 just over 100 TOMY have been initiated and become fully operational. From 2019, when the neoliberal government of New Democracy came to power, until now only six new TOMY were launched.

Third, although the significance of school nursing is recognized by school actors in Greece, the professionals are restricted to the daily, scheduled care of children and adolescents with special needs and emergency care, as a systematic review of relative legislative data through a span of three decades (1982–2011) reveals [25]. Although a decade has passed since the above review, little if any progress has been achieved and equally little evidence has been generated in the field. As confirmed in a recent work, school nurses are typically employed in special education schools at the request of parents of children with special healthcare needs after a relevant medical diagnosis has been issued by a public hospital doctor [26]. Given the above, two conclusions are extracted: (a) the duties of school nurses have not been expanded to cover the health needs of the total school population and (b) school nurse presence in schools remains sporadic and limited to the needs of children with special needs.

Fourth, school heath education as a subject is delivered within the context of the Greek educational system, basically in primary and secondary education, on a non-compulsory and voluntary basis. As a field of knowledge, within applied educational practice, it may adopt two distinct forms of curricula implementation: either as a cross-thematic subject within the compulsory timetable or as an extracurricular school activity, outside teaching hours. However, the constant modifications in its pedagogical methods and its place in the school curricula led to further marginalization of health education in schools [16,22].

Fifth, and in relation to the wider sociopolitical context, it is important to emphasize that health services and the public educational system, as facets of the social welfare state, suffered the effects a of a decade (2009–2018) of harsh austerity measures imposed by Greek governments in return loans from the International Monetary Fund, the European Union and the European Central Bank (widely acknowledged as the “Troika”) following a debt crisis, which contributed to the deregulation and deterioration of labor, standards of living and the welfare system. The prolonged financial crisis and the imposition of austerity measures resulted in a reduction in health expenditures, the widening inequalities in access and downgrade of the quality of services [27].

Taking a critical public health and critical pedagogy point of view, we aim to produce a radical policy proposal informed by the principles of equality, equity, democracy, inclusion and grassroots participation decisions, following an ecological model framework which advocates embracing ecological approaches, political economic theory and critical pedagogy [28], and arguing against the de-politicization of public health [29]. At the heart of this theoretical model lies empowerment education which ensures community participation and dialogue at a personal level but also in social arenas to reinforce the individuals’ and communities’ ability to gain control over their own health.

## 3. Research Methods

This scoping review aimed to explore and systematically map the school health services and school health education curriculum in Greek primary, secondary and higher education, seeking to identify key problems, drawbacks and disadvantages in policy implementation. The objective, therefore, of this work was to locate certain gaps in the current literature and legislation as for the above-mentioned fields of action and then to highlight the gaps that need to be covered by policymaking, adopting a problem-solving perspective. This purpose is served by synthesizing the evidence thematically. The search strategy utilized the following categories: school health services, school health education curricula, school nursing, integration and critical pedagogy, all with reference exclusively to Greece, since this country is taken as a case study. The search followed two stages, one search with English and one with Greek key-terms. Irrespectively of the language employed, the terms remained identical, i.e., “health education & Greece” “school health & Greece”, “school health services & Greece”, “school health education & Greece”, “health programs & school & Greece”, “health interventions & school & Greece”, etc.

### 3.1. Research Questions

The following research questions provided the framework of this scoping review:To what extent and how school health services are offered in Greek primary and secondary education?To what extent and how school education is included in the curricula of Greek primary and secondary education?To what extent is health education included in the medical schools’ curricula?To what extent and how is school nursing included in schools?

### 3.2. Inclusion and Exclusion Criteria

A set of inclusion criteria was utilized from the outset of this work in order to exclude a multitude of irrelevant findings.

First, as this review aimed to trace evolution in policies, documents that focused merely on experiments, interventions and randomized-control trials were excluded.

Second, the time of the accumulated documents covered a period of two decades, i.e., from 2003 to 2023, thus allowing enough space to observe the evolution of the two fields of knowledge and action (i.e., school health services and school health curricula in primary and secondary education), whilst as for the syllabuses of the medical schools we scrutinized only the current ones (i.e., 2023). Hence, documents before 2003 have been excluded.

Third, the spectrum of the genres of the collected documents ranged from publications in journals (original research, reviews or editorials), syllabuses of medical schools, doctoral theses, conference proceedings and policy texts extracted from the official websites of the MoH and MoE, whilst publications in mass media or other sources were excluded.

Fourth, relevant data extracted from the corpus of documents by locating key terms in them were then classified into the above categories, thus building a narrative synthesis. For instance, the data had to be necessarily relevant to Greece, as this was the case studied.

### 3.3. Findings

During the first round of our scoping review, 282 documents were initially identified by searching the above-mentioned databases that fit the set of specific inclusion criteria. Subsequently, the duplicates were removed and the titles and abstracts of 202 have been scrutinized for further consideration. Out of 202 documents screened, six have not been accessible in full text and have consequently been excluded as well as 12 more documents due to the exclusion criteria above. During the second round, 184 full text records were assessed for eligibility, excluding 22 documents that did not meet all of the inclusion criteria. The third round ended up with 162 documents that have been finally included in this scoping review, consisting of seven doctoral theses, 14 legislative texts, 27 conference proceeding, 107 publications in peer reviewed journals and seven syllabuses of tertiary education. From the 162 documents, only 17 correspond to the set of research questions.

In what follows (Table 1) a detailed process of search in databases is depicted. All the following search terms included also “Greece” or “Greek” to limit the data to our case study.

### 3.4. The CriSHEP Plan

This article attempts to fill in this gap by presenting an innovative radical integrated policy plan bringing together school health services with health education curricula embedded in critical public health and critical pedagogy. The fundamental axes of a Critical School Health and Education Policy (CriSHEP) is presented below (Figure 1).

On-site provision of health services at school, adopting the settings approach of health promotion, will be organized and implemented by the PPH units in deprived urban areas, giving priority to pupils from lower social strata and disadvantaged regions;The on-site provision of health services includes early detection, treatment and monitoring of health problems of children and adolescents, covering almost the entire age range of the minor population. Organization of medical and school nursing services, which will provide on-site screening tests to check: (a) visual acuity and color blindness, (b) oral hygiene, (c) diseases of the spine (e.g., scoliosis), (d) vaccination coverage and infectious diseases, (e) development (body measurements—weight, height, head circumference), (f) checking vaccination coverage, (g) carrying out Mantoux tuberculosis reactions, (h) carrying out vaccinations;Design of school health education curricula based on the principles of critical pedagogy, oriented to equality, social justice and solidarity, and aiming to re-conceptualize health as a social good and not an individual responsibility. In this frame, the whole-school approach is adopted in order to engage the entire school community, i.e., students, parents, teachers at all levels of basic education (preschool education, primary education, secondary education), thus enhancing participation towards an ecological model;Ensuring provision of school meals to all primary schools of the country and including special school meals for pupils with special nutritional needs;Pre-service teachers’ education: Rejuvenation of health education curricula in primary and preschool education departments of tertiary education based on critical pedagogy;Pre-service medical doctors’ education: Rejuvenation of health education curricula in medical schools of tertiary education based on critical pedagogy.

## 4. Discussion

In summary, this article reviewed the evolution of school health services and health education curricula in Greece over a span of two decades, aiming to produce thereafter am integrated and radical policy plan, based on a vision of certain principles and theoretically underpinned by critical public health and critical pedagogy paradigms.

First, it is found that school health services are rarely based in a permanent and sustainable mode in a school setting, and they are rarely accessible for the whole school population. In contrast, SHS constitute a non-compulsory duty of PHC and as such operate in a sporadic and fragmented mode in primary and secondary education delivered basically by school nurses recruited for children with special condition.

Second, it is observed that the vast majority (reaching 75%) of the studies under consideration in terms of the scoping review focus on single interventions or health-related topics, disregarding the contextual factors that impede or promote the enactment of curricula. The political context of the policies implemented in practice is dismissed from studies exploring or surveying SHS and SHE, arguably because of the allegedly neutral public-health domain [29].

Third, SHE as a subject does not retain a permanent position in school curricula; it has operated as an extracurricular and cross-curricular activity [30,31]. Moreover, SHE operates under the medical hegemony retaining a bio-medical and individualistic character focusing on personal health and disease prevention [16], whereas schoolteachers lack training, experience and leadership from the health education officers [32]. In addition, the optional character of SHE in the school curricula is identified as a barrier to its implementation [31]. In respect to school nursing, it is restricted to the daily, scheduled care of children and adolescents with special needs and emergency care. Finally, it is noted that none of the seven medical schools in Greece have health education as an autonomous module, although several relevant subjects, such as public health and preventive medicine, include elements of health education and health literacy [25,26].

In line with a recent large observational study conducted in 30 EU countries, this review emphasizes the need to adapt to the contemporary and perplexed health priorities of pupils and extend SHS beyond traditional screening or vaccination procedures [15]. The two distinctive executive bodies of Ministry of Health and Ministry of Education need to establish a sustainable collaboration aiming to integrate SHS with SHE into a robust health and education policy which will protect and foster the real needs of the new generation in the post pandemic world.

The lack of school-based health services in conjunction with the absence of a primary healthcare system, at least recently, increases pupils’ unmet health needs, especially for those from lower social strata. By contrast the development of a concrete and comprehensive integrated system of school health is expected to fill in the gaps. From this perspective, based upon the deficiencies identified in this review, we devised and propose the Critical School Health and Education Policy (CriSHEP).

The CriSHEP is embedded in critical public health and critical pedagogy paradigms, in that it interrogates the current status, it views public health as political endeavor and not a neutral field of science and it seeks to bring social justice and solidarity. Within this framework, it is anticipated that the universal health coverage, initiated in the major reform of PHC in 2018, will be expanded and deepened through the reform of school health services and unmet health needs, as exemplified in the OECD’s (2021) “Health at a glance” edition [21], signaling that inequalities in access to health services due to structural and economic barriers will be reduced.

## 5. Conclusions

The imperative for the post-pandemic world and future generations is to revisit and reinvent public health policies that apply in the school setting. The two cardinal components of what constitutes school health, that is health services on the one hand and health education on the other, need to be integrated. However, the new vision of public health policies needs to address the disastrous effects of the humanitarian crisis and the lessons stemming from this. In this light, the problem-solving perspective that produces evidence towards a new egalitarian and integrated perspective in school health and health education is at stake at a time of crises.

## Figures and Tables

**Figure 1 healthcare-11-01678-f001:**
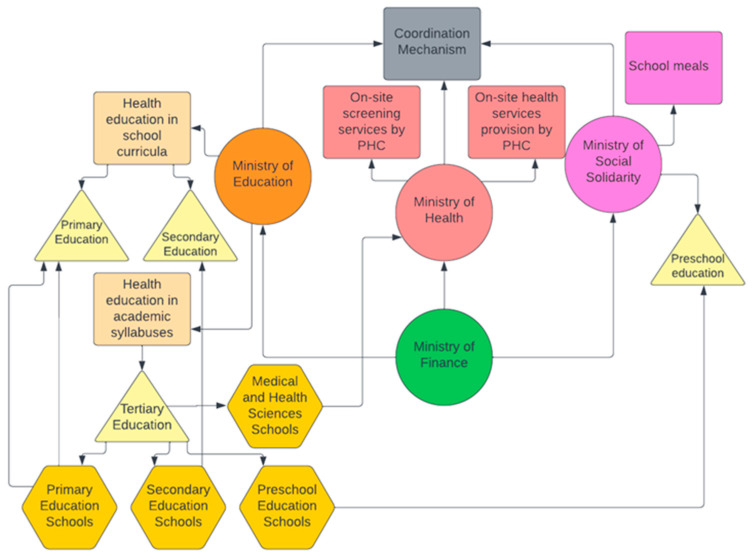
Critical School Health and Education Policy (CriSHEP).

**Table 1 healthcare-11-01678-t001:** Scoping Review Selection Process.

Key Concepts	Search Terms	Time Span	Genres
School health	Health services primary school	2003–2023	Academic publications Legislation
	Health services secondary education		
	Health services preschool education		
	TOMY		
	Health Centers		
	Primary Healthcare Service		
School health education	Health education primary school	2003–2023	Academic publications Legislation
	Health education secondary school		
	School health education		
Health Education	Health education	2023	Syllabuses of Medical Schools
	Health literacy		

## Data Availability

Not applicable.

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
