# Peer review of "School Health Services and Health Education Curricula in Greece: Scoping Review and Policy Plan"

_healthcare, 2023, doi:10.3390/healthcare11121678_

Round 1

Reviewer 1 Report

Dear Authors,

this is an interesting work. Please find here my comments.

1.       Abstract: I am not sure that this statements “This article serves a twofold purpose: a) it provides a genealogy of school health services  and school health education over a span of two decades, highlighting the major problems and policy gaps by taking Greece as a case study, b) it offers subsequently a robust and radical policy plan to rejuvenate and to integrate school health services and school heath education curricula” actually sustains the aim of the scoping review and they could results in overstating the work. My advice is to keep the rational focused on the need for a scoping review aimed at […].

2.       Introduction: Establishing why integrating school health services and health education is important for highlighting the worth of this area, e.g., consider providing a brief introduction to the significance of this topic and its relevance to public health or education.

3.       The subheading findings should be “selection process”

4.       Discussion: It would be beneficial to expand on the practical implications of implementing the CriSHEP. Consider discussing how the integration of school health services and the adoption of critical pedagogy principles would benefit students, schools, and the broader community. It is important to acknowledge any limitations of the study and potential challenges in implementing the proposed policy plan. Discussing these limitations can provide a balanced perspective and open avenues for further research and improvement

5.       Other: please use the PRISMA SCr to self-assess the work and cite it in the methods http://www.prisma-statement.org/Extensions/ScopingReviews?AspxAutoDetectCookieSupport=1

Author Response

  1. We edited the abstract to make more explicit the meanings and to remain focused on the aims of the scoping review.
  2. We included a brief introduction to the significance of this topic and its relevance to public health or education.
  3. We changed the subsection heading to “selection process”.
  4. We revised the Discussion section accordingly,  elaborating on on the practical implications of implementing the CriSHEP. 
  5. We used the PRISMA SCr to self-assess the work and cite it in the methods 

Reviewer 2 Report

The document represents an interesting topic, however, I suggest that its structure, functional logic and systematization be reviewed; the main observations are:

The abstract must be strengthened from its logic, clarity and coherence; the situation under study must be evidenced to a greater degree, objectives, results, conclusions...

“genealogy of school health services and school health education over a span of two decades” What are the dimensions of said analysis? What is the scope of genealogy?

“offers subsequently a robust and radical policy plan to rejuvenate and to integrate school health services and school heath education curricula” take care of the use of adjectives that are subject to multiple interpretations, as well as terms that must be better contextualized.

The methodological component is not clear, the approach, method and main techniques used must be specified, the inclusion criteria of the units of analysis must be specified, it must be specified whether it is a systematic review or which technique was used.

“in a sporadic, limited and fragmented mode” What are the implications of these terms, they lend themselves to multiple interpretations, multiple meanings.

“health education retain a constantly changing position in school curricula” changes as to what? specify.

“this article is the first” Review this type of statement, what is its foundation

In the abstract: complement the article's contributions to the advancement of knowledge of the object).

Likewise, some conclusive idea should be pointed out.  it is important to structure it from a scientific position where the type of article and the situation under study, the contexts involved in the research, the methodological components, the results, the main conclusions, are more clearly perceived. adjust.

In the introduction: Some constructs such as school health services and school health education curricula, health policies, approach of health promotion and public health, Sisyphean regression of health education to medical hegemony, among others must be discussed and argued from the contrast with the review of the literature, evidencing to what extent they form gaps or problematizing axes

…critical pedagogy and critical public health (theoretical and practical implications of these approaches).

…the rejuvenation, reinforcement and integration of school health and school health education (argue these ideas, review terminology, contrast from the supporting theory).

…concurrently, school health education, at least in a span of the last three decades, retains a marginal and constantly changing position in school curricula (review these statements, on which they are based, strengthen arguments, do not take the ideas for granted).

… interdisciplinary unit, extracurricular school activity (argue these ideas)

“capitalist crises” (argue these ideas).

In the introduction, the objectives of the article, research questions should be evidenced. It is important to infer how this research contributes to the advancement of science, its contributions and social impact, identifying direct and indirect beneficiaries, as well as the relevance of this research within the framework of emerging paradigms that explain the subject studied.

The theoretical and empirical gaps of the situation under study should be evidenced to a greater degree, as well as the descriptors associated with the main variables studied; the variables and their dimensions should be described with greater emphasis. It is necessary to strengthen the citations of recent scientific literature that allow contrasting the descriptors of reality and show the state of knowledge related to the situation under study.

The situation under study, as well as the associated descriptors, the importance and relevance of the topic, the sense of contextualization in the region studied must be strengthened in their description; Likewise, the citation system of recent scientific literature related to the subject should be strengthened (this aspect is really too weak)

It is important to propose a deductive route; it is necessary that the implications of the problem situation be described on a macro, meso and micro plane; situation that must be described in greater depth because it must be better argued; the problem must start from a more general scope, before falling into the variables of the investigation; that is, they should begin by presenting descriptors of the macro; then locate yourself in the meso and micro plane. It is necessary to conceptualize each of the dimensions of the variables worked on, not limiting yourself to identifying them, but explaining and arguing how they are being conceptualized in the scope of your article.

The review of the literature should show the sequentiality of ideas between paragraphs from one section to another to guarantee internal coherence, sudden ruptures are perceived between one section and another.

The review of the literature must show a critical and argumentative apparatus based on the main constructs: Health policies in the context of the country studied, how these policies are configured, the interdisciplinary vision between health and pedagogy, school health services, educational policies,, among others must be discussed and argued from the contrast with the review of the literature, correspondence with objective lops, these constructs represent central axes of sequentiality and discussion in this section.

It is important to review the statement of each section in the literature review in such a way that a sense of totality and completeness is evidenced with the theoretical system that it represents.

It is necessary to strengthen the citation system of recent scientific literature that allows contrasting the theoretical postulates presented.

The titles of each section or section must show a greater scope in correspondence with the theoretical relations and contents treated, for example, the section called. vision needs to be checked.

In the methodology, is necessary to identify and justify the typology of the article.  Is important specify the predominant research approach, as well as the type of design. The methodological component is not clear, the approach, method and main techniques used must be specified, the inclusion criteria of the units of analysis must be specified, it must be specified whether it is a systematic review or which technique was used.  Likewise, the information gathering techniques and instruments must be specified; instruments for each of the identified samples; criteria of validity and reliability of the instruments. Likewise, it is necessary to specify the information processing techniques to be obtained. The procedural systematization must be reviewed organize the research stages, in correspondence with the objectives of the article, type of research according to knowledge to be produced and expected products. Provide greater evidence of the operationalization of its variables, main dimensions and indicators that allow measuring the behavior of its fundamental variables.

It is recommended to prepare a table that accounts for the data related to the types of sources consulted, for example, if they are doctoral theses, data inherent to the themes, lines of research, axes addressed; If the indexing data of the journals are articles, in short, the profile of the analysis units must be strengthened for a better understanding of the reader.

It is important to argue the degree of representativeness of the analysis worked in correspondence with the sense of totality and completeness of the investigated object, possibilities that the results can be generalized.

Organize a results and discussion section where it is considered a better correspondence with the procedural systematization that is necessary to declare in the methodology, that is, how each component of the design leads to the different sections that are presented in the results. Review the correspondence between the methodological systematization and the systematization of the results, attend to variables, dimensions and indicators and their operationalization.

The discussion section should be strengthened. Organize the results in such a way that they are evident in relation to the objectives; there is no evidence of contrast between the objectives - supporting theory - meaning of the data itself - argument of the researchers.

It is necessary to strengthen the citations of recent scientific literature, citations of contrasting literature must be updated.

A section of conclusions should be presented where is it shown correspondence with the identified objectives; demonstrate possibilities of generalization of the research to contexts with similar characteristics. Conclusions must transcend results.

Author Response

Following  the Reviewer’s 2 comments, we: 

-revised totally the Abstract to respond to all of the reviewer's comments. The new version is authored again in order to be more explicit and structured. Although we have not included subheadings in the Introduction, it is evident now that is structured as follows: a) introduction, b) objectives, c) results and d) conclusions.

-have re-authored the whole of the Introduction to incorporate the comment regarding the deductive route from macro to meso and then to micro level of analysis and critical argumentation, to eliminate subjectivity (i.e. robust and rejuvenate) and misinterpretations, to avoid broad unsupported statements (e.g. genealogy), to explain in greater detail our arguments to the international audience (not taking them as granted), to review the literature showing now a critical and argumentative apparatus based on the main constructs.

-have edited the Methodology section to make more explicit the methods used in terms of this scoping review.

-have re-organized and strengthened the Discussion to reflect the research questions framework and adjust to the findings of this work with more relevant and up-to-date literature.

-have added a Conclusion section.

Round 2

Reviewer 1 Report

my previous comments have been addressed

Reviewer 2 Report

A cordial greeting.

The adjusted version of the article presents very good argumentation and coherence.

Thank you for applying the requested settings.